# Peri-Surgical Inflammatory Profile Associated with Mini-Invasive or Standard Open Lumbar Interbody Fusion Approaches

**DOI:** 10.3390/jcm10143128

**Published:** 2021-07-15

**Authors:** Giovanni Lombardi, Pedro Berjano, Riccardo Cecchinato, Francesco Langella, Silvia Perego, Veronica Sansoni, Fulvio Tartara, Pietro Regazzoni, Claudio Lamartina

**Affiliations:** 1Laboratory of Experimental Biochemistry and Molecular Biology, IRCCS Istituto Ortopedico Galeazzi, 20161 Milan, Italy; giovanni.lombardi@grupposandonato.it (G.L.); silvia.perego@grupposandonato.it (S.P.); veronica.sansoni@grupposandonato.it (V.S.); 2Department of Athletics, Strength and Conditioning, Poznań University of Physical Education and Sport, 61-871 Poznań, Poland; 3OU GSpine 4, IRCCS Istituto Ortopedico Galeazzi, 20161 Milan, Italy; pberjano@gmail.com (P.B.); dott.cecchinato@gmail.com (R.C.); c.lamartina@chirurgiavertebrale.net (C.L.); 4IRCCS Istituto Neurologico Nazionale C. Mondino, 27100 Pavia, Italy; tartarafulvio@gmail.com; 5Department of Trauma Surgery, University Hospital Basel, 4031 Basel, Switzerland; p_regazzoni@bluewin.ch

**Keywords:** minimally invasive surgery, cytokine, inflammatory mediators, post-surgery recovery, spine fusion, bone, clinical outcome, surgical outcome, interbody fusion

## Abstract

Background: Different surgical approaches are available for lumbar interbody fusion (LIF) to treat disc degeneration. However, a quantification of their invasiveness is lacking, and the definition of minimally invasive surgery (MIS) has not been biochemically detailed. We aimed at characterizing the inflammatory, hematological, and clinical peri-surgical responses to different LIF techniques. Methods: 68 healthy subjects affected by single-level discopathy (L3 to S1) were addressed to MIS, anterior (ALIF, *n* = 21) or lateral (LLIF, *n* = 23), and conventional approaches, transforaminal (TLIF, *n* = 24), based on the preoperative clinical assessment. Venous blood samples were taken 24 h before the surgery and 24 and 72 h after surgery to assess a wide panel of inflammatory and hematological markers. Results: martial (serum iron and transferrin) and pro-angiogenic profiles (MMP-2, TWEAK) were improved in ALIF and LLIF compared to TLIF, while the acute phase response (C-reactive protein, sCD163) was enhanced in LLIF. Conclusions: MIS procedures (ALIF and LLIF) associated with a reduced incidence of post-operative anemic status, faster recovery, and enhanced pro-angiogenic stimuli compared with TLIF. LLIF associated with an earlier activation of innate immune mechanisms than ALIF and TLIF. The trend of the inflammation markers confirms that the theoretically defined mini-invasive procedures behave as such.

## 1. Introduction

Minimally invasive surgery (MIS) techniques rapidly gained attention in surgical practice thanks to the significant advantage in reducing tissue exposure and trauma. In the last few years, MIS indications have grown, now covering the whole set of surgeries [1,2]. In spine surgery, MIS approaches cover the whole range of procedures [3,4]. The technological advances acquired have allowed the application of different MIS procedures to a wide range of conditions (e.g., fractures, deformities, tumors) [1], but the more common use of MIS is in interbody fusion for the treatment of back pain [5], and sagittal [6] and coronal malalignment correction [7]. Hence, along with the potential benefits for patients, the reliability of MIS procedures in spine surgery relies on the pros and cons. Inclusion criteria for MIS are similar, and somehow wider, compared to conventional open surgeries [3]. Furthermore, several advantages have been claimed: limited muscle trauma, quicker post-operative recovery and rehabilitation, reduced hospital stay, and improved clinical outcomes [1,8]. Spinal interbody fusion is the elective treatment in case of disc degeneration, spinal instability, trauma, and deformity. Anatomically, the “body” is the biggest vertebra portion and the greatest area for fusion and, biomechanically, the majority of the load goes through the vertebral body. For this reason, the interbody fusion represents a pivotal point to obtain high fusion rates and good clinical outcomes. Several approaches are used for lumbar interbody fusion (LIF) in MIS (anterior (ALIF), lateral (LLIF), or eXtreme-lateral (XLIF) [4]) and conventional open procedures (posterior (PLIF) and transforaminal (TLIF)).

Unlike conventional procedures, LLIF and ALIF reach the target spine level(s) via tubular access or through blunt muscle dilation rather than wide dissection of paraspinal tissues. However, data about the real biological advantages of these types of surgeries are still lacking [9]. It has been proposed that MIS procedures may result in a lesser degree of muscle injury, pain, blood loss, lower complication rate (e.g., lower rate of infections), and fewer use of hospital resources compared to conventional surgeries [6,10,11]. The main disadvantages are: the high technical demands, challenging learning/specific skills, and high operative costs due to the use of intraoperative imaging/guidance and expensive equipment [1].

From a clinical point of view, a quantification of spine arthrodesis invasiveness is still lacking. According to a recent review, C-reactive protein (CRP) and creatine kinase (CK) resulted as the most reliable markers of invasiveness [1]. However, being highly non-specific markers, their real utility, in this context, remains limited [8]. Thereby, the aim of this study was to characterize the inflammatory, hematological, and clinical peri-surgical responses to MIS (ALIF and LLIF) and open conventional approaches (TLIF) in a cohort of otherwise healthy patients, only affected by single-level lumbar discopathy.

## 2. Materials and Methods

### 2.1. Study Design and Setting

This is a single-center cohort, prospective observational study. From June 2012 to June 2015, 68 consecutive subjects affected by single-level discopathy (L3 to S1) were recruited in the same institution. The protocol followed the institutional review board (IRB, PQ 7.5.125, version 4, dated 22 January 2015, approved by the IRCCS Istituto Ortopedico Galeazzi) approval, and was performed in accordance with the Helsinki Declaration.

### 2.2. Participants

All subjects were healthy, only affected by discopathy with indication to surgical treatment. Inclusion criteria were: age > 18 years, L3–S1 single-level discopathy, indication for LIF, no actual or previous chronic pathological conditions, and regular intraoperative and post-surgery course. Exclusion criteria were tumor, infection, or any medical condition capable to influence the inflammatory markers. Patients underwent to routine clinical evaluations and interventions, and the study protocol did not affect the clinical and surgical procedure. All patients were informed about the risks and benefits of the procedures and the use of personal data in aggregated form. They all provided the written informed consent and agreed to undergo the surgery in the presence of the recruiting physician.

Blood samplings were performed according to the clinical standards, and those analytes not routinely determined were assayed in plasma remaining from routine analysis. Venous blood was taken, in standard conditions, 24 h before the surgery (T0), and 24 h (T1) and 72 h (T2) after surgery, in SST™II Advance serum tubes for clinical chemistry evaluation and K2EDTA tubes for hematology and inflammatory status profiling (BD vacutainer, Becton Dickinson & Co., Franklin Lakes, NJ, USA). Pre-analytical phase guidelines for venous blood drawing and sample processing and storing [12,13] were closely followed.

Patients were addressed to ALIF (*n* = 21), TLIF (*n* = 24), or LLIF (*n* = 23) based on the clinical indication and preoperative collegial discussion. The age at the time of surgery, body mass index (BMI), American Society of Anesthesiologists (ASA) grade, surgery length, duration of hospitalization, transfusion requirements, and disposition at the time of discharge were collected for all patients. Objective measurements of serum levels of inflammatory cytokines and a marker of muscle damage were compared between these three cohorts. Although the study was not a randomized trial, the patient demographics were analyzed to ensure similar cohorts.

### 2.3. Surgery Description

The posterior transforaminal approach, TLIF, formerly our standard open control group, was performed with the patients in prone position and under general anesthesia. A posterior midline incision (from 5 to 10 cm) was followed by paravertebral bilateral muscle dissection. The disc space was reached via a unilateral laminectomy and inferior facetectomy. After the annulotomy and endplate preparation with the disc removal, the trialing, and finally the insertion of the cage, for interbody fusion were performed. Furthermore, the posterior procedure was completed with 4 pedicle screws (two below and two above the disc treated) and 2 rods’ placement.

The ALIF stand-alone procedures were performed with the patients in the supine position, under general anesthesia. The Pfannenstiel approach provided a left retroperitoneal corridor to the disk space. After a transverse median incision, the rectus fascia was opened, and the peritoneum was exposed. The fat pad and the peritoneum were medially moved to identify the left psoas muscle. Upon reaching the bifurcation of great vessels, the promontorium was exposed. The disc was removed, and the cage was placed after an accurate endplate preparation. One plate and four screws were placed to increase the primary implant stability.

LLIF was used to reach a lateral direct retroperitoneal approach to the disc space. The procedures were completed without posterior supplementary surgeries. The surgery was performed by approaching the lumbar vertebral bodies with a 90° lateral pathway (eXtreme lateral—XLIF) through the psoas muscle with the patient lying in a left or right sided position. A small (4–6 cm) lateral incision was performed prior to fluoroscopic control. A psoas muscle splitting with progressively larger dilators was achieved with the use of real-time neuro-monitoring. Upon reaching the lateral aspect of the anterior column, the retractor was placed to visualize the disc space, allowing the disc removal and cage implant.

In our MIS cases (LLIF and ALIF), no additional posterior surgeries were performed.

### 2.4. Biochemical and Hematological Characterization

Hematological analysis included: white (WBC 10^9^/L) and red blood cell count (RBC 10^12^/L), hemoglobin concentration ((Hb), g/L), hematocrit (Ht, %), mean corpuscular volume (MCV, fL), mean hemoglobin content (MCH, pg), mean corpuscular hemoglobin content (MCHC, g/L), platelet count (Plt, 10^9^/L), and relative (%) and absolute (10^9^/L) leucocyte formulas (neutrophils-Neu, lymphocytes-Ly, Monocytes-Mo, Eosinophils-Eo, Basophils-Ba), performed on a XT 1800i (Sysmex, Corp., Kobe, Japan), and the imprecision was <1.5%. Internal quality controls were performed through e-Check (LE Tri level, Sysmex, Kobe, Japan) and instrumentation was controlled by external proficiency testing.

Serum was assayed for the indexes of organ damage (alkaline phosphatase (ALP), U/L; aspartate aminotransferase (AST), U/L; alanine aminotransferase (ALT), U/L: enzymatic method), martial status (iron: ferene method; ferritin, transferrin: chemiluminescent method), and systemic inflammation (CRP: turbidimetric method) on an Architect i4000SR (Abbott, Lake Bluff, IL, USA). The instrument was routinely checked. A panel of 37 inflammatory biomarkers, comprising non-specific markers of inflammation (sCD163, interleukin (IL)-8, IL-11, IL-12p40, IL-12p70, IL-27, pentraxin (PTX)-3), TNF/TNFR members (a proliferation-inducing ligand (APRIL)/tumor necrosis factor superfamily (TNFSF)-13, B-cell activating factor (BAFF)/TNFSF13B, LIGHT/TNFSF14, soluble TNF receptor (sTNFR)-1, sTNFR-2, sCD30, TNF-related weak inducer of apoptosis (TWEAK)/TNFSF-12), B- and T-cell cytokines (IL-2, IL-32, IL-34, IL-35, TSLP), cytokines receptors and co-receptors (gp130, sIL-6R), class II cytokines (IL-10, IL-19, IL-20, IL-22, IL26, IL-28/interferon (IFN)λ2, IL-29/IFNλ1, IFNα2, IFNβ, IFNγ), matrix metalloproteinases (MMPs) and tissue remodeling factors (MMP-1, MMP-2, MMP-3, chitinase-3-like protein 1 (CHI3L1)/YKL-40), and bone turnover markers (osteocalcin (OC), osteopontin (OPN)) have been assayed through a high-sensitivity multiplex bead-based immunofluorescent assay (Bio-Plex Pro™ Human Inflammation Panel 1, 37-Plex, Bio-Rad Laboratories, Inc., Hercules, CA, USA) within the same plasma sample aliquot at the same time, on a MagPix™ System (Bio-Rad Laboratories, Inc.), based on manufacturer specifications. All samples were tested in duplicate.

### 2.5. Statistical Analysis

Statistical analysis was conducted with SPSS software (IBM, Armonk, NY, USA). Sample size was calculated using plasma concentrations of IL-8, an inflammatory marker of tissue damage, as a primary endpoint index, by assuming a two-tailed type I α error of 0.05, a power (1-β error probability) >0.99, a standard deviation of 4.5 pg/mL, and a between-group difference of 4 pg/mL [8]. Based on these assumptions, the effect size (f) is 1.52, from which a minimum sample size of 21 subjects per group was deduced. With the sample size of each group being <30, non-parametric statistics were applied. Data are described as median and range. The within-group time-dependent changes were analyzed by Friedman’s test and the post hoc Dunn’s multiple comparison test. The between-group differences at each time-point were evaluated by the unpaired Mann–Whitney’s test. Correlation analysis was performed by Spearman’s test (0.25 < r ≤ 0.50: fair correlation, 0.50 < r ≤ 0.75: good correlation, r > 0.75: strong correlation). Significance was set at *p* < 0.05.

## 3. Results

### 3.1. Patients

We enrolled 68 patients (39 (57.3%) women), mean (SD) age: 46.3 (10.2) years. Surgery lasted, on average, 127 min (35–439 min). Two patients were rated as ASA 3, and the remaining were ASA 2 and 1 (27 and 39, respectively). Of all the patients, 21 underwent ALIF (30.9%), 24 TLIF (35.3%), and 23 LLIF (33.8%). In the TLIF group, 12 patients (50%) were operated on at L4–L5 and 12 patients (50%) at L5–S1. All ALIF cases were performed at L5–S1. Six LLIF cases were performed at L3–L4 (26.1%) and seventeen at L4–L5 (73.9%). A lateral plate was placed to provide additional stability in 4 cases. Otherwise, 20 patients underwent the stand-alone procedure, 21 patients were approached from the left, and 2 from the right side.

### 3.2. Hematological and Clinical Chemistry Profile

Compared to T0, all surgeries consistently decreased RBC, Hb, and Ht (*p* < 0.001) more in TLIF and less in ALIF, while not significantly in LLIF. WBC increased at T1, mainly in ALIF (*p* < 0.001), and only in LLIF did the increase continue at T2 (*p* < 0.01) (Table 1, Figure 1A). Compared to ALIF, TLIF experienced a greater decrease in RBC (*p* < 0.01) and Ht (*p* < 0.05) at T1. At the same time-point, LLIF experienced lower Neu% (*p* < 0.05) and higher Ly% (*p* < 0.05) and Ba% (*p* < 0.05) (Table 1, Figure 1B). In terms of amplitude of changes, compared to ALIF, the T1–T0 and T2–T0 differences for RBC, Hb, and Ht were much lower in TLIF (*p* < 0.001), while they were comparable in LLIF. On the contrary, the T2–T0 Δ for Ht was increased in LLIF (*p* < 0.05), indicating an early recovery (Figure 1C). In TLIF, the T1–T2 Δ for Plt (*p* < 0.05) and the T2–T0 Δ for Ly% (*p* < 0.05) were decreased, while in LLIF, the T2–T1 Δ for WBC was higher than in ALIF (*p* < 0.05).

Compared to T0, serum iron was importantly reduced in the entire cohort at T1 (*p* < 0.001) and T2 (*p* < 0.001), as well as in the single approaches (all, *p* < 0.001). Transferrin was decreased at T1 (*p* < 0.001) and T2 (*p* < 0.001) in TLIF, and this, consequently, drove the decreasing trend in the entire cohort (*p* < 0.01 and *p* < 0.001). Ferritin showed a significant increase only in the whole cohort at T2 (*p* < 0.01). ALP was reduced at T1 and T2 in TLIF (*p* < 0.01). AST and ALT were globally unaffected, but the AST-to-ALT ratio resulted increased at T1 and T2 in the whole cohort (*p* < 0.05) (Table 2, Figure 1A).

Compared to ALIF, in TLIF, there was a lower level of transferrin at T2 (*p* < 0.01), whilst higher serum activity of AST was recorded for both TLIF and LLIF at T1 (*p* < 0.01, *p* < 0.05) and T2 (*p* < 0.01) (Table 2, Figure 1B).

Compared to ALIF, in TLIF, the T1–T0 Δ for iron (*p* < 0.05), transferrin (*p* < 0.01), ALP (*p* < 0.01), and ALT (*p* < 0.05), and the T2–T0 Δ for transferrin (*p* < 0.05) and ALP (*p* < 0.05) were significantly lower, while the T2–T0 Δ for AST-to-ALT ratio was higher (*p* < 0.05). On the contrary, the T2–T0 Δ for iron (*p* < 0.05), transferrin (*p* < 0.05), AST (*p* < 0.05), and AST-to-ALT ratio (*p* < 0.05) were all higher (Figure 1C).

### 3.3. Inflammatory Mediators Profile

#### 3.3.1. Non-Specific Markers of Systemic Inflammation

Compared to T0, CRP was strongly increased at T1 and T2 in all the approaches (*p* < 0.001). Only in TLIF did the increase at T1 not reach significance. sCD163 resulted increased in the entire cohort at T2 (*p* < 0.01), while it was slightly decreased at T1 in ALIF (*p* < 0.01) and TLIF (*p* < 0.001), and more so at T2 in TLIF (*p* < 0.001). IL-11 was increased in the entire cohort at T2 (*p* < 0.01), whilst, by analyzing the single approaches, it was very slightly increased in TLIF at T1 (*p* < 0.05). PTX-3 was strongly induced in all the approaches (*p* < 0.001 vs. T1; *p* < 0.05 vs. T2) (Table 3, Figure 2A).

Compared to ALIF, none of the acute phase cytokines differed. Only CRP at T1 in LLIF was much higher (*p* < 0.05) (Table 3, Figure 2B).

Considering the amplitude of changes compared to ALIF, the T1–T0 and T2–T0 Δ for CRP in LLIF were higher (*p* < 0.05). In TLIF, the T2–T0 Δ for sCD163 was also decreased (*p* < 0.01). The T2–T1 Δ for IL-11 (*p* < 0.05), IL-12p40 (*p* < 0.05), IL-12p70 (*p* < 0.01), and IL-27 (*p* < 0.05) were all increased in LLIF. The same difference was increased in TLIF for IL-11 and IL-12p40 (*p* < 0.05), whilst the T2-T0 Δ for IL-12p40 was decreased (*p* < 0.05). The amplitude of changes for IL-8 and PX-3 were not modified by the approach (Figure 2C).

#### 3.3.2. Members of the TNF/TNFR Superfamily

Compared to T0, increased APRIL/TNFSF13 was recorded at T1 in ALIF and TLIF (*p* < 0.05). BAFF/TNFSF13B was increased at T1 only in the entire cohort (*p* < 0.05), while LIGHT/TNFSF14 was strongly increased in the whole cohort at T2 and in TLIF at T1 (*p* < 0.01). sTNFR-1 and sTNFR-2 were importantly increased in the entire population at both T1 and T2 (*p* < 0.001). Such an increase was replicated for sTNFR-1 in TLIF at T1 (*p* < 0.01) and T2 (*p* < 0.05) and in LLIF at T1 (*p* < 0.05), and for sTNFR-2 in ALIF at T2 (*p* < 0.001), and in TLIF and LLIF at T1 (*p* < 0.01 and *p* < 0.05, respectively) and T2 (*p* < 0.01 and *p* < 0.05, respectively). sCD30 was unchanged. TWEAK/TNFSF12, instead, was consistently decreased at T2 in the whole cohort (*p* < 0.001), as well as in ALIF at T2 (*p* < 0.001) and in LLIF at T1 (*p* < 0.05) and T2 (*p* < 0.001). In TLIF, an increase at T2 (*p* < 0.001) was recorded (Table 3, Figure 2A).

Compared to ALIF, no differences were found at any time-point in APRIL/TNFSF13, BAFF/TNFSF13B, LIGHT/TNFSF14, and sTNFR-2. sTNFR-1 was increased in both TLIF and LLIF at T1 (*p* < 0.05). sCD30 was increased at T1 in TLIF and at T2 in LLIF (*p* < 0.05), while TWEAK, at T2, was decreased in TLIF (*p* < 0.05) and increased in LLIF (*p* < 0.01) (Table 3, Figure 2B).

The analysis of the amplitude of changes indicated that, compared to ALIF, the main differences were in TLIF, in which decreases were recorded in APRIL/TNFSF13 and sCD30 (T2–T1, *p* < 0.05), and LIGHT/TNFSF14 and TWEAK (T2–T0, *p* < 0.05); the T1–T0 Δ for sTNFR-1, instead, was strongly increased (*p* < 0.05). For LLIF, only the T2–T0 Δ for sTNFR-2 increased (*p* < 0.05), by about 50% (Figure 2C).

#### 3.3.3. B- and T-Cell Cytokines

In general terms, humoral immunity was not involved in the peri-surgical period, as demonstrated by the slight changes in the subset of B- and T-cell-derived cytokines. Compared to T0, a slight decrease was found for IL-2 and IL-35 in LLIF at T1, while, at the same time-point, IL-35 resulted slightly increased in ALIF (*p* < 0.05). TSLP was decreased, instead, in TLIF at both T1 and T2 (*p* < 0.05) (Table 3, Figure 2A).

Compared to ALIF, IL-2 was slightly decreased in both TLIF (*p* < 0.01) and LLIF (*p* < 0.05) at T1. In LLIF, higher levels of IL-32 at T2 and IL-34 at T1 (*p* < 0.05) were also recorded (Table 3, Figure 2B).

The T1–T0 Δ for IL-2 was lower in both LLIF and TLIF (*p* < 0.05), compared to ALIF. In LLIF, however, the T2–T1 Δ was strongly increased (*p* < 0.05). An overlapping situation was seen for IL-35. Il-34 was higher in TLIF in the T1–T0 Δ (*p* < 0.05). TSLP, instead, was decreased over the T0–T2 observation, compared to ALIF, although increased in T2–T1 Δ (*p* < 0.05) (Figure 2C).

#### 3.3.4. Cytokine Receptors and Co-Receptors

Compared to T0, gp130 resulted moderately decreased at T2 in the entire cohort (*p* < 0.05), as well as in ALIF (*p* < 0.05) and TLIF (*p* < 0.001), and additionally, although slighter, for sIL-6Rβ (*p* < 0.05 and *p* < 0.001, respectively) (Table 3, Figure 2A).

Consequently, compared to ALIF, the gp130 concentrations were significantly higher in LLIF at both T1 and T2 (*p* < 0.01) (Table 3, Figure 2B).

The amplitude of changes of the T2–T1 Δ for gp130 (*p* < 0.05) and sIL-6Rβ (*p* < 0.001), however, were importantly reduced in TLIF compared to ALIF (Table 3, Figure 2C).

#### 3.3.5. Class II Cytokines

Compared to T0, IL-10 (*p* < 0.01) and IL-26 (*p* < 0.05) were strongly increased in the whole cohort and in all the approaches. IL-19, IFNα, and IFNγ were decreased only at T1 in LLIF, IL-20 was decreased in the entire cohort at T1 (*p* < 0.01) and T2 (*p* < 0.05) and in LLIF at T1 (*p* < 0.05), and IL-22 was decreased in ALIF at T1 (*p* < 0.05) (Table 3, Figure 2A).

Compared to ALIF, however, no significant differences were found in TLIF and LLIF. Only IL-28 was evidently lower in TLIF and LLIF at T2 (*p* < 0.01), as were IL-29 at T1 in LLIF (*p* < 0.05), and IFNα2 at T1 in both TLIF and LLIF (*p* < 0.05) (Table 3, Figure 2B).

Compared to those recorded in ALIF, the amplitude of changes between the different time-points were similar for IL-10 and IL-29. In LLIF, IL-19 (*p* < 0.01), IL-26 (*p* < 0.05), IFNα2 (*p* < 0.05), and IFNγ (*p* < 0.05) were all lower in the T1–T0 Δ and higher in the T2–T1 Δ. IL-22 was lower in both comparisons (*p* < 0.01, *p* < 0.05), while the T2–T0 Δ for IFNβ was higher (*p* < 0.05). In TLIF, instead, the T1–T0 Δ for IL-22 (*p* < 0.05) and the T2–T0 Δ for IL-28 (*p* < 0.05) were significantly lower than in ALIF (Figure 2C).

#### 3.3.6. Matrix Metalloproteinases and Tissue Remodeling Factors

Compared to T0, a T2-associated decrease in MMP-2 was found in the entire cohort (*p* < 0.01) and in TLIF (*p* < 0.001). At T1, MMP-3 was slightly increased in ALIF (*p* < 0.001) and decreased in TLIF (*p* < 0.05). CHI3L1 was equally strongly increased in all the conditions (Table 3, Figure 2A).

Compared to ALIF, there was only a T2-associated consistent decrease in MMP-2 (*p* < 0.01) (Table 3, Figure 2B), which was reflected in a lower T2–T0 Δ (*p* < 0.01) compared to ALIF (Figure 2C).

#### 3.3.7. Bone Markers

Interestingly, OC and OPN were importantly affected by the surgery. Compared to T0, while a general, and somehow sustained, decrease in OC was observed, a strong increase in OPN was found, with the only exception of LLIF at T1 (Table 3, Figure 2A).

Compared to ALIF, OC resulted consistently decreased at T2 in LLIF (*p* < 0.05), while OPN was slightly but significantly increased at T1 in both TLIF and LLIF (*p* < 0.05) (Table 3, Figure 2B).

The amplitude of changes between the time-points were not different for OPN, while lower T2–T0 Δ (*p* < 0.05) and T2–T1 Δ (*p* < 0.01) were recorded for OC (Figure 2C).

Time and surgical approach-dependent changes in the most relevant markers, i.e., those whose statistically significant changes could be potentially related to a clinically relevant outcome (transferrin, AST, CRP, IL-2, gp130, TWEAK/TNFSF12, MMP2, OC, OPN), are presented in Figure 3.

## 4. Discussion

For the first time, here, we provided an overview of the inflammatory changes that take place within the peri-surgical period in a cohort of carefully selected patients who underwent LIF surgeries with three different approaches. With the effect of surgery (standard and MIS) on the classical cytokines (e.g., IL-1, IL-6, TNFα) being known, in this study, we analyzed the acute changes in the profile of a wide panel of “second line” mediators, responsible for the complex network of tissue responses to the injury and activating healing and function recovery. It was previously highlighted that peri-surgical changes of CRP, IL-6, and IL-10 were the most likely, discriminating between standard and MIS procedures in LIF [1,14]. However, these markers do not predict the injury-elicited healing response.

In this study, it firstly emerged that the three approaches elicit similar responses, with the main differences attributable to the hematological and martial compartment, innate immunity response, early remodeling response, and bone metabolic activation.

### 4.1. Hematological Profile

The higher blood loss and transfusion rate in open surgery versus MIS is still debated in the literature. Conventionally, the posterior approach (TLIF) is by definition related to high muscle trauma and bleeding when compared to ALIF and LLIF stand-alone procedures. As expected, due to blood loss, the RBC, Hb, and Ht are mostly influenced by the muscle injury. However, while TLIF caused the greater decrease in these parameters, in LLIF, such a time-dependent decrease is absent. Moreover, the amplitudes of changes between the different time-points in ALIF and LLIF were comparable. This implies that: (i) blood losses are reduced in LLIF (and ALIF) compared to TLIF and (ii) a greater recovery potential is available in patients who underwent LLIF approaches (and ALIF) if compared to TLIF. Notably, LLIF also featured a greater early post-surgical increase of WBC, compared to the other approaches, possibly linked to an enhanced innate immunity activation.

### 4.2. Clinical Chemistry Profile

As a direct consequence of blood losses, iron loss was greater in TLIF and lower in LLIF compared with ALIF. Notably, serum transferrin concentrations, which were unaffected in ALIF and LLIF during the post-surgery phase, were reduced (up to 50%) in TLIF. During the observation, transferrin levels remained much higher in LLIF compared to ALIF (up to 50%). The transforaminal approach decreased ALP activity levels in the postsurgical phase and also compared with the other procedures. The higher degree of technical invasiveness of the TLIF approach [15] is marked by the decrease in ALP [15]. According to Chikhani et al. [16], an acute and reversible decrease in liver function (e.g., marked by ALP and transaminases) is associated with the prone position. Thereby, in the TLIF approach, the prone position may also influence AST and ALT responses. The reduced liver function seems to be confirmed by significantly lower ALT activity levels in TLIF in the T0–T1 interval. On the contrary, AST activity levels were increased in both TLIF and LLIF in the post-surgery phase, compared to ALIF. This means that the skeletal muscle involvement [17,18] in these two approaches is greater than in the anterior approach. Accounting for the intrinsically non-specific nature of these markers, on the one hand, we may justify the greatest hepatic distress in prone (TLIF) and lateral (LLIF) decubitus, and on the other hand, the greater muscular trauma (LLIF and TLIF) if compared to ALIF.

### 4.3. Inflammatory Mediators Profile

Notably, although CRP increased in all the approaches, the greatest rise was in LLIF, with the absolute higher CRP concentration recorded at T1. CRP is a member of the pentraxins (together with PTX-3), a class of innate immunity-derived antibody-like proteins involved in acute phase reaction [19]. The rise in LLIF was linked to a greater post-surgical relative level of other innate immunity-derived cytokines. IL-11, IL-12p40, IL-12p40, and IL-27 are all produced by macrophages and are implied in T-cell activation. Further, IL-2, involved in the autocrine activation of T-cells [20], and IL-35, produced by T-regulatory cells (Treg) [21], increased in the post-surgery phase in LLIF, while they decreased in TLIF, compared to ALIF. sCD30, IL-32, and gp130 increased in LLIF, at T2, compared to ALIF. sCD30 is expressed by B- and T-cells and activates NF-κB, the key pro-inflammatory transcription factor. However, this action is addressed in increasing the T helper CD4+ compartment, which has regulatory function on the immune system and in limiting the proliferation of the T cytotoxic CD8+ population [22]. IL-32 is produced by T-cells and natural killer (NK) cells, following IL-2-dependent activation, and induces the expression of TNFα, IL-6, and other pro-inflammatory cytokines in macrophages, and it supports osteoclast differentiation from circulating monocytes [23]. gp130 is a co-receptor for IL-6 and many other cytokines, thus it is necessary for the expression of the pro-inflammatory phenotype [24]. The largest increases of these cytokines in LLIF could be read as an activation of the regulatory compartment of the immune system, as an attempt in limiting the inflammatory process.

TWEAK (TNFSF12) is a TNF superfamily member, with overlapping functions with TNF, produced by monocytes, dendritic cells (DC), and NK, promoting inflammation and wound healing. TWEAK also positively regulates angiogenesis by promoting proliferation and migration of endothelial cells [24]. TWEAK is decreased in all the approaches; however, compared to ALIF, in LLIF, its concentrations remained consistently higher, while in TLIF, the T0–T2 decline was more pronounced. MMP-2 (gelatinase A) degrades collagen type IV and is involved in the tissue remodeling processes in health and diseases. Similar to TWEAK, MMP has pro-angiogenic roles [25] and it was decreased in TLIF at T2, compared to ALIF and LLIF, and also the T0–T2 trend was decreasing in TLIF.

### 4.4. Bone Turnover Profile

Surgery affected bone function. OPN, a cytokine connecting the immune system to the bone metabolism, also involved in the onset and progression of spine deformities [25], was increased in all the procedures, but, at T1, it was relatively higher in TLIF and LLIF, compared to ALIF. Contrarily, OC, the bone-derived hormone linking bone and energy metabolisms [26], was consistently decreased over the observation, although much more in LLIF at T2. OC is considered a marker of bone formation and its decrease in LLIF could be linked to the rise of IL-32. From an energy metabolism point of view, the decrease in OC could signal a reduced energy demand [27,28].

By considering all the approaches, the main evidences are: (i) onset of a slight anemic state due to blood loss, (ii) activation of innate immune mechanisms with only a slight or no activation of the humoral response, (iii) sudden activation of tissue remodeling and angiogenesis, and (iv) activation of bone turnover and osteo-immune crosstalk.

By comparing the approaches in the post-surgery period, some important differences were found: (i) compared to TLIF, the martial status was improved in ALIF and LLIF, with higher levels of iron and transferrin and comparable serum concentrations of ferritin, (ii) the acute inflammatory phase response (CRP, sCD163) was higher in LLIF than in ALIF, while the lowest was in TLIF, (iii) LLIF and ALIF displayed an improved profile of pro-angiogenic factors (MMP-2, TWEAK) compared to TLIF, and (iv) changes in bone markers levels (OC and OP) may be related to the different actions on vertebral bones associated with the approach or, eventually, to a different metabolic effort consequent to the inflammatory response.

Despite the significant results of our study, some biases must be considered. Notably, the elevated degree of homogeneity of the study cohort is supported by the fact that at baseline, no difference was found in any of the studied analytes. Besides the interest around the peri-surgical response, the choice of such a limited period of observation (96 h in total) is based on the fact that by the third day post-surgery, patients who underwent LIF surgery are discharged from the hospital. This means that all variables which have been kept under control during hospitalization (e.g., diet, physiotherapy, medications) are lost when the patient is sent home. This could have an important impact on such a panel of highly variable parameters; indeed, the inflammatory, immune, and organ function responses are not exhausted within this period and a longer investigation would have provided additional information, useful to define the entire response. Another limitation of the study is related to the evaluation of the post-operative results and the surgeon that performed the procedure (P.B., C.L., or F.T.). The effectiveness of a “standard open” or minimally invasive procedure should also be evaluated in terms of clinical and radiographic outcomes not included in our study. Mainly, to consolidate the acquired knowledge and investigate the effects of perioperative blood markers on long-term outcomes, such as clinical and radiographic outcomes, further high-quality studies and longer follow-up are mandatory.

## 5. Conclusions

Surgeons define invasiveness according to different criteria, such as incision length, duration of recovery, hematic blood loss, pain, and muscle damage. In spine surgery, the identification of the minimally invasive criteria is challenged by the different types of surgical procedures and therapeutic aims, as well as by the different selection of patients. Few studies have investigated the role of inflammatory mediators in vertebral surgery. The aim of our study was first to compare the biochemical criteria of invasiveness in procedures defined as “standard open” and “mini-invasive”. Furthermore, we characterized the inflammatory, hematological, and clinical peri-surgical responses to different LIF approaches (MIS: ALIF and LLIF, open conventional: TLIF) in a cohort of otherwise healthy patients, only affected by single-level lumbar discopathy. The four areas of investigation have been deepened. MIS procedures (ALIF and LLIF) demonstrated a lower incidence of post-operative anemic status when compared with the conventional posterior approach. The MIS procedures have allowed for a faster recovery of values in terms of the humoral body’s early response. Especially, the LLIF procedure seems to activate the early innate immune mechanisms significantly faster when compared to ALIF and TLIF. Furthermore, in terms of early angiogenesis activators, the MIS procedures have shown an improved profile with respect to the conventional TLIF procedures. No clear data are available regarding bone marker responses. In our study, the OC, the bone-derived hormone, was consistently decreased over the observation, although much more in LLIF. This last factor seems to be influenced by other mediators of the inflammatory response. As there is not a clearer interpretation, more detailed studies seem necessary to investigate this aspect. The results of our study did not provide evidence of the superiority of the MIS over conventional approaches or vice-versa, and further studies are needed to understand how blood markers affect clinical outcomes.

## Figures and Tables

**Figure 1 jcm-10-03128-f001:**
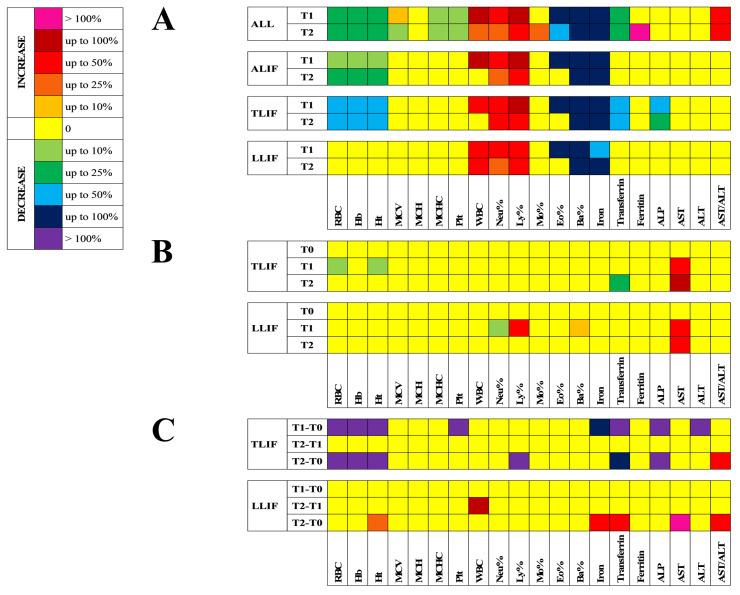
Hematological parameters and clinical chemistry profile. (**A**) Comparison of the hematological and clinical chemistry profiles with the relative baseline value (T0) in the whole cohort (ALL), and in the three approaches (ALIF, TLIF, and LLIF). (**B**) Comparison of the hematological and clinical chemistry profiles with ALIF, at each time-point. (**C**) Comparison of the amplitude of changes, between the different time-points (T1–T0, T2–T1, T2–T0) of the hematological and clinical chemistry profiles with those of ALIF. The color scale indicates the degree of percentage variation. ALIF: Anterior Lumbar Interbody Fusion; TLIF: Transforaminal Lumbar Interbody Fusion; LLIF: Lateral Lumbar Interbody Fusion; RBC: red blood cells; Hb: hemoglobin; Ht: hematocrit; MCV: mean corpuscular volume; MCH: mean hemoglobin content; MCHC: mean corpuscular hemoglobin content; Plt: platelet count; WBC: white blood cells; Neu: neutrophils; Ly: lymphocytes; Mo: monocytes; Eo: eosinophils; Ba: basophils; ALP: alkaline phosphatase; AST: aspartate aminotransferase; ALT: alanine aminotransferase.

**Figure 2 jcm-10-03128-f002:**
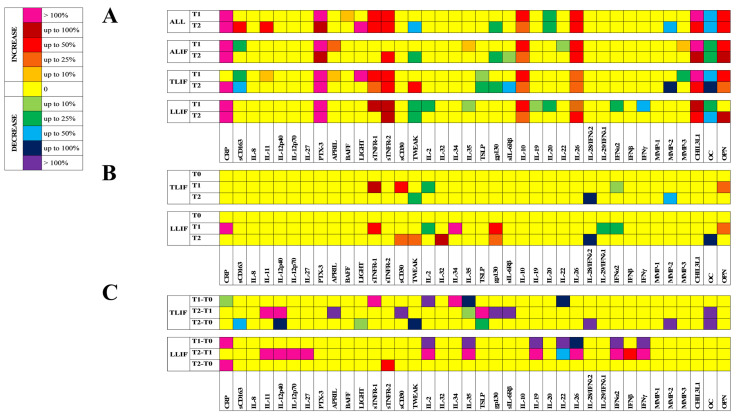
Inflammatory markers, cytokines, cytokine receptors and co-receptors, and tissue remodeling factors. (**A**) Comparison of the inflammatory mediators’ profiles with the relative baseline value (T0) in the whole cohort (ALL), and in the three approaches (ALIF, TLIF, and LLIF). (**B**) Comparison of the inflammatory mediators’ profiles with ALIF, at each time-point. (**C**) Comparison of the amplitude of changes, between the different time-points (T1–T0, T2–T1, T2–T0) of the inflammatory mediators’ profiles with those of ALIF. The color scale indicates the degree of percentage variation. ALIF: Anterior Lumbar Interbody Fusion; TLIF: Transforaminal Lumbar Interbody Fusion; LLIF: Lateral Lumbar Interbody Fusion; CRP: C-reactive protein; sCD: soluble cluster of differentiation; IL: interleukin; PTX-3: pentraxin-3; APRIL: a proliferation-inducing ligand; BAFF: B-cell activating factor; LIGHT: homologous to lymphotoxin, exhibits inducible expression and competes with HSV glycoprotein D for binding to herpesvirus entry mediator, a receptor expressed on T lymphocytes; sTNFR: soluble tumor necrosis factor receptor; TWEAK: tumor necrosis factor-related weak inducer of apoptosis; TSLP: thymic stromal lymphopoietin; gp130: glycoprotein 130; sIL-6Rβ: soluble interleukin 6 receptor β; IFN: interferon; MMP: matrix metalloproteinases; CHI3L1: chitinase-3-like protein 1; OC: osteocalcin; OPN: osteopontin.

**Figure 3 jcm-10-03128-f003:**
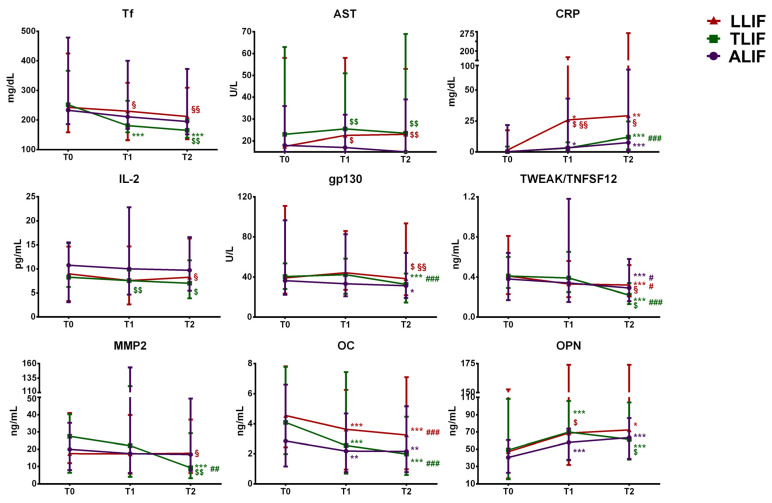
Time-dependent changes in the most clinically relevant markers in the different surgical approaches. The figure shows the median value and the range for each time-point. *: difference vs. T0 (*: *p* < 0.05; **: *p* < 0.01; ***: *p* < 0.001); #: difference vs. T1 (#: *p* < 0.05; ##: *p* < 0.01; ###: *p* < 0.001); $: difference vs. ALIF ($: *p* < 0.05; $$: *p* < 0.01); §: difference vs. TLIF (§: *p* < 0.05; §§: *p* < 0.01). Tf: transferrin; AST: aspartate aminotransferase; CRP: C-reactive protein; IL-2: interleukin 2; gp130: glycoprotein 130; TWEAK/TNFSF12: tumor necrosis factor-related weak inducer of apoptosis/tumor necrosis factor-related soluble factor 12; MMP2: matrix metalloproteinases 2; OC: osteocalcin; OPN: osteopontin.

**Table 1 jcm-10-03128-t001:** Hematological profile over the peri-surgical period.

	Surgical Approach											
	ALL			ALIF			TLIF			LLIF		
	T0	T1	T2	T0	T1	T2	T0	T1	T2	T0	T1	T2
RBC (10^12^/L)	4.70 (3.42–6.33)	4.27 (2.91–5.35) ***	3.96 (2.77–5.31) ***, ###	4.59 (3.85–5.68)	4.30 (3.30–5.01) *	3.96 (2.82–4.77) *	4.88 (4.15–6.33)	3.90 (2.91–4.74) ***, $$	3.43 (2.77–4.73) ***	4.51 (3.42–6.02)	4.33 (3.24–5.35) §§§	4.22 (3.08–5.31) §§§
Hb (g/L)	13.6 (9.6–17.8)	12.1 (7.9–16.6) ***	11.5 (8.0–16.2) ***, ###	13.5 (9.6–17.8)	12.3 (9.1–15.2) *	11.6 (8.3–14.1) ***	14.9 (11.1–16.8)	11.4 (7.9–14.2) ***	10.3 (8.0–14.0) ***	13.5 (10.3–17.2)	12.3 (8.9–16.6) §	11.9 (8.6–16.2) §§
Ht (%)	40.8 (30.3–49.8)	36.5 (25.7–45.6) ***	34.3 (24.5–46.3) ***, ###	40.4 (32.1–49.8)	36.5 (30.2–42.8) *	34.4 (25.6–41.0) ***	42.9 (35.0–48.0)	33.3 (25.7–41.0) ***, $	30.3 (24.5–40.5) ***	39.9 (30.3–45.1)	39.1 (27.2–45.6) §§	37.3 (26.6–46.3) §§
MCV (fL)	88.6 (61.0–99.6)	88.8 (60.5–100.3) ***	88.4 (61.8–101.6) ***	87.3 (67.9–98.0)	87.70 (69.1–98.1)	87.9 (68.2–97.4)	89.3 (64.3–99.6)	89.6 (64.6–100.3)	89.0 (64.6–101.6)	88.6 (61.0–96.0)	88.8 (60.5–96.7)	88.4 (61.8–97.6)
MCH (pg)	29.6 (19.9–36.6)	29.8 (19.8–35.6)	29.7 (19.9–34.9)	29.4 (20.3–33.1)	29.6 (20.7–33.4)	29.6 (20.4–32.9)	30.4 (20.2–34.3)	30.0 (19.8–33.5)	29.8 (20.0–34.3)	29.9 (19.9–36.6)	30.1 (20.1–35.6)	30.1 (19.9–34.9)
MCHC (g/L)	33.8 (29.9–38.1)	33.5 (29.9–36.9) ***	33.4 (29.8–35.8) ***	33.8 (29.9–36.6)	33.7 (29.9–36.0)	33.5 (29.8–35.1)	33.8 (31.4–35.9)	33.3 (30.7–35.4)	32.9 (30.9–35.4)	34.1 (32.5–38.1)	33.7 (31.9–36.9)	33.5 (32.2–35.8)
Plt (10^9^/L)	228 (104–462)	215 (120–440) ***	207 (113–458) ***, ##	226 (139–462)	223 (137–440)	208 (138–458)	220 (132–334)	209 (121–285)	194 (113–377)	233 (104–381)	215 (120–304))	212 (132–294)
WBC (10^9^/L)	6.81 (3.48–12.9)	10.28 (6.26–17.91) ***	8.63 (3.49–15.93) ***, ###	7.07 (3.59–12.9)	12.20 (6.73–17.91) ***	7.78 (5.07–15.93) #	7.17 (4.22–10.47)	10.66 (6.27–15.39) ***	9.06 (4.87–10.71) #	6.63 (3.48–10.74)	9.21 (6.26–15.45) ***	8.90 (3.49–13.63) **
Neu (%)	57.3 (27.1–83.8)	76.3 (52.8–90.4) ***	68.6 (50.2–91.2) *** ###	61.05 (43.4–83.8)	79.2 (64.8–90.4) ***	69.7 (50.2–81.1) *, ##	53.9 (33.1–76.0)	76.0 (60.0–90.3) ***	70.0 (52.0–81.9) ***	58.1 (27.1–69.2)	73.9 (52.8–86.9) ***, $	64.4 (53.5–91.2) ***
Ly (%)	33.1 (11.7–50.6))	16.1 (4.4–36.1) ***	20.5 (5.5–36.1) ***, ###	28.5 (11.7–46.0)	12.9 (4.7–24.4) ***	19.0 (11.4–35.3) **, ##	35.4 (17.9–47.6)	15.1 (4.4–31.4) ***	19.6 (12.4–35.9) ***	32.3 (22.7–50.6)	18.1 (7.5–36.1) ***, $	24.1 (5.5–36.1) ***
Mo (%)	7.6 (3.8–13.5)	7.7 (3.0–12.6)	8.5 (3.3–13.5) *	7.6 (4.3–11.0)	8.4 (3.0–12.4)	8.2 (5.9–13.1)	7.7 (4.9–13.5)	8.0 (5.3–11.6)	9.0 (5.6–13.5)	7.6 (3.8–12.8)	7.5 (3.3–12.6)	8.4 (3.3–12.7)
Eo (%)	1.9 (0.1–14.20)	0.1 (0.0–3.8) ***	1.3 (0.0–7.7) *, ###	1.3 (0.1–7.6)	0.1 (0.0–3.4) **	1.3 (0.0–4.3) #	2.0 (0.3–14.2)	0.1 (0.0–1.4) **	1.3 (0.0–7.7)	1.9 (0.7–9.7)	0.7 (0.0–3.8) **, §	1.6 (0.0–5.8)
Ba (%)	0.3 (0.0–1.6)	0.1 (0.0–0.5) ***	0.1 (0.0–0.4) ***	0.3 (0.0–1.6)	0.1 (0.0–0.4) **	0.1 (0.0–0.3) *	0.3 (0.1–1.1)	0.1 (0.0–0.3) ***	0.1 (0.0–0.4) *	0.4 (0.2–0.7)	0.1 (0.0–0.5) ***, $, §	0.1 (0.0–0.4) ***

Values are expressed as median (range). *: difference vs. T0 (*: *p* < 0.05; **: *p* < 0.01; ***: *p* < 0.001). #: difference vs. T1 (#: *p* < 0.05; ##: *p* < 0.01; ###: *p* < 0.001). $: difference vs. ALIF ($: *p* < 0.05; $$: *p* < 0.01). §: difference vs. TLIF (§: *p* < 0.05; §§: *p* < 0.01; §§§: *p* < 0.001). RBC: red blood cells; Hb: hemoglobin; Ht: hematocrit; MCV: mean corpuscular volume; MCH: mean corpuscular hemoglobin; MCHC: mean corpuscular hemoglobin content; Plt: platelets; WBC: white blood cells; Neu: neutrophils; Ly: lymphocytes; Mo: macrophages; Eo: eosinophils; Ba: basophils, ALIF: Anterior Lumbar Interbody Fusion; TLIF: Transforaminal Lumbar Interbody Fusion; LLIF: Lateral Lumbar Interbody Fusion.

**Table 2 jcm-10-03128-t002:** Clinical chemistry parameters over the peri-surgical period.

	Surgical Approach											
	ALL			ALIF			TLIF			LLIF		
	T0	T1	T2	T0	T1	T2	T0	T1	T2	T0	T1	T2
Iron (μg/dL))	99.0 (29.0–226.0)	40.0 (13.0–113.0) ***	23.0 (7.0–106.0) ***	103.0 (29.0–176.0)	40.0 (17.0–113.0) ***	23.0 (11.0–87.0) ***	108.0 (36.0–226.0)	30.0 (13.0–60.0) ***	16.0 (8.0–64.0) ***	83.5 (29.0–174.0)	45.0 (22.0–94.0) ***	36.0 (7.0–106.0) ***, §
Transferrin (mg/dL)	245 (158–479)	210 (132–400) **	194 (135–375) ***	233 (186–479)	211 (170–400)	195 (151–373)	252 (186–367)	181 (158–265) ***	165 (140–200) ***, $$	243.5 (158–425)	230.5 (132–326) §	212 (135–309) §§§
Ferritin (ng/mL)	62.1 (2.0–666.0)	127.0 (7.0–858.0)	126.8 (9.0–1000) *	60.80 (2.0–501.0)	117.9 (7.0–718.0)	121.0 (9.0–1000.0)	67.0 (7.0–666.0)	128.0 (23.0–858.0)	174.0 (66.0–915.0)	64.4 (5.0–465.8)	114.0 (9.0–795.2)	135.5 (11.0–711.4)
ALP (U/L)	64.5 (29.0–105.0)	53.5 (28.0–109.0)	53.5 (25.0–119.0)	48.0 (41.0–105)	46.5 (30.0–109.0)	49.0 (25.0–119.0)	69.0 (40.0–104.0)	48.0 (28.0–74.0) **	53.0 (35.0–70.0) **	61.0 (29.0–103.0)	60.0 (30.0–105.0)	61.0 (31.0–97.0)
AST (U/L)	18.0 (9.0–63.0)	19.5 (10.0–58.0)	21.0 (9.0–69.0)	18.0 (9.0–36.0)	17.0 (10.0–32.0)	13.5 (11.0–39.0)	23.0 (10.0–63.0)	25.5 (12.0–51.0) $$	23.5 (9.0–69.0) $$	17.5 (11.0–58.0)	22.5 (12.0–58.0) $	23.0 (12.0–53.0) $$
ALT (U/L)	17.5 (5.0–116)	16.0 (6.0–89.0)	16.0 (6.0–87.0)	16.0 (8.0–49.0)	12.5 (6.0–41.0)	12.0 (6.0–47.0)	19.0 (5.0–83.0)	16.5 (9.0–66.0)	16.0 (6.0–43.0)	19.0 (9.0–116.0)	19.5 (9.0–89.0)	18.0 (9.0–87.0)
AST/ALT	0.9 (0.4–3.9)	1.2 (0.6–2.7) *	1.3 (0.5–3.2) *	1.0 (0.6–1.8)	1.1 (0.6–2.2)	1.1 (0.5–2.2)	0.9 (0.4–2.6)	1.3 (0.7–2.7)	1.3 (0.7–3.2)	0.9 (0.4–3.9)	1.2 (0.7–2.3)	1.3 (0.5–2.3)

Values are expressed as median (range). *: difference vs. T0 (*: *p* < 0.05; **: *p* < 0.01; ***: *p* < 0.001). $: difference vs. ALIF ($: *p* < 0.05; $$: *p* < 0.01). §: difference vs. TLIF (§: *p* < 0.05; §§§: *p* < 0.001). ALP: alkaline phosphatase, AST: aspartate aminotransferase; ALT: alanine aminotransferase.

**Table 3 jcm-10-03128-t003:** Inflammatory markers over the peri-surgical period.

	Surgical Approach											
	ALL			ALIF			TLIF			LLIF		
	T0	T1	T2	T0	T1	T2	T0	T1	T2	T0	T1	T2
**Acute phase**												
CRP (mg/dL)	0.50 (0.02–21.71)	5.21 (0.46–208.5) ***	17.72 (0.45–271.4) ***, ###	0.15 (0.02–21.71)	3.31 (0.46–43.00) *	7.50 (0.45–66.67) ***	0.20 (0.03–4.29)	3.09 (1.16–7.76)	11.86 (2.0–24.64) ***, ###	1.465 (0.03–17.45)	25.84 (1.09–208.5) *, $, §§	29.30 (1.01–271.4) **, §
sCD163 (ng/mL)	84.75 (26.65–299.49)	80.82 (27.49–610.36)	63.31 (18.99–458.94) ***, ##	74.65 (32.05–255.90)	67.13 (32.11–411-27) **	57.96 (23.80–187-48)	97.96 (44.39–152-49)	84.73 (18.99–86.33) ***	58.86 (18.99–86.33) ***, ###	78.11 (26.65–299.49)	84.00 (26.65–610.36)	79.59 (26.34–458.94) §
IL-8 (pg/mL)	42.65 (16.75–76.84)	42.65 (16.22–119.80)	37.73 (8.70–67.51)	46.61 (16.75–76-84)	43.28 (22.64–69.15)	39.77 (16.75–61.38) #	36.88 (22.64–53.75)	36.46 (16.75–56.75)	33.14 (8.70–64.22)	46.12 (22.64–73.05)	42.65 (16.22–119.8)	42.04 (16.22–67.51)
IL-11 (pg/mL)	0.62 (0.05–3.22)	0.62 (0.05–3.22)	0.89 (0.07–6.16) **, #	0.53 (0.05–2.38)	0.62 (0.05–2.83)	0.65 (0.08–2.32)	0.76 (0.07–1.76)	0.80 (0.05–2.85) *	0.90 (0.07–3.44)	0.57 (0.07–3.22)	0.50 (0.05–3.22)	1.05 (0.09–6.16)
IL-12p40 (pg/mL)	53.08 (16.00–129.70)	50.48 (16.00–190.70)	51.02 (13.47–144.30)	63.64 (30.53–91.10)	57.44 (36.14–190.70)	51.02 (31.02–91.10)	49.94 (36.51–75.37)	45.49 (25.80–91.02)	48.68 (20.97–127.50)	52.74 (27.40–96.00)	46.24 (18.12–96.00)	55.05 (19.94–144.30)
IL-12p70 (pg/mL)	2.41 (0.10–9.82)	2.09 (0.10–6.70)	2.33 (0.10–6.31)	3.18 (0.10–5.53)	3.21 (1.21–6.70)	2.73 (0.10–3.98)	2.55 (0.71–4.21)	2.10 (1.77–4.76)	2.41 (0.90–3.69)	2.41 (1.13–9.82)	2.25 (0.81–4.76)	2.41 (1.45–6.31) #
IL-27 (pg/mL)	13.82 (0.10–42.46)	10.44 (0.10–46.38)	13.82 (0.10–65.63)	5.53 (0.10–39.49)	5.53 (0.10–46.38)	13.82 (0.10–39.49)	13.82 (0.10–39.06)	14.24 (0.10–32.18)	13.82 (0.10–45.83)	14.24 (0.10–42.46)	10.44 (0.10–45.83)	17.90 (0.10–65.63) #
PTX-3 (ng/mL)	0.62 (0.15–8.12)	2.22 (0.19–11.73) ***	1.15 (0.13–5.78) *, ###	0.60 (0.34–2.94)	2.05 (0.62–5.70) ***	1.08 (0.58–4.13) *, ##	0.50 (0.22–0.80)	2.99 (0.66–6.05) ***	1.09 (0.60–3.24) ***, ###	0.74 (0.15–2.65)	2.23 (0.19–11.73) **	1.50 (0.13–5.78) *
**TNF superfamily**												
APRIL/TNFSF13 (ng/mL)	52.63 (12.98–137.73)	57.43 (76.48–161.60)	56.16 (8.95–109.40)	50.10 (12.98–94.83)	55.65 (7.65–99.83) *	55.86 (30.05–92.12)	57.28 (30.05–92.12)	62.21 (44.27–128.39) *	52.20 (32.11–109.40) #	48.39 (21.54–109.76)	51.45 (14.20–127.76)	56.84 (30.86–108.90)
BAFF/TNFSF13B (ng/mL)	7.86 (3.72–30.06)	8.52 (3.90–23.18) *	5.99 (3.27–16.74) ##	6.48 (3.72–23.85)	7.27 (4.16–21.73)	6.85 (4.16–14.79)	7.86 (5.85–11.90)	8.57 (4.74–12.60)	8.23 (3.90–13.19)	8.05 (4.59–30.06)	9.94 (3.90–23.18)	7.99 (3.85–16.74)
LIGHT/TNFSF14 (pg/mL)	1.73 (0–257.4)	3.53 (1.5–146.1)	15.12 (1.5–277.4) **, #	1.73 (1.73–257.40)	9.83 (1.73–61.47)	11.44 (1.73–185.30)	1.73 (1.73–127.20)	20.86 (1.73–50.41) **	46.95 (1.73–277.40)	3.58 (1.50–146.30)	27.82 (1.73–146.10)	56.02 (1.73–253.30)
sTNFR-1 (ng/mL)	2.19 (0.91–7.53)	3.16 (1.34–12.02) ***	2.97 (1.26–9.48) ***, #	1.95 (0.91–7.53)	2.38 (1.34–5.21)	2.43 (1.26–5.34)	2.55 (1.62–5.30)	3.64 (1.88–8.46) **, $	3.14 (1.61–7.73) *	2.13 (1.13–5.23)	3.36 (1.98–12.02) *, $	3.12 (1.62–9.48)
sTNFR-2 (ng/mL)	3.99 (1.70–9.85)	5.73 (2.03–30.63) ***	5.70 (2.41–26.93) ***	3.67 (2.10–9.13)	5.39 (2.03–13.89) ***	5.19 (3.18–10.80)	4.08 (1.70–9.10)	5.43 (2.06–19.06) **	5.69 (2.62–12-25) **	3.49 (1.82–9.85)	6.55 (2.08–30.63) *	6.79 (2.14–26.93) *
sCD30 (ng/mL)	0.39 (0.16–1.28)	0.42 (0.18–1.40)	0.39 (0.16–1.49)	0.36 (0.16–1.02)	0.35 (0.18–0.98)	0.37 (0.16–0.74)	0.42 (0.27–0.87)	0.45 (0.23–0.97) $	0.39 (0.22–0.62)	0.40 (0.22–1.28)	0.49 (0.26–1.40)	0.46 (0.27–1.49) $, §
TWEAK/TNFSF12 (ng/mL)	0.40 (0.17–0.81)	0.37 (0.15–1.18)	0.27 (0.13–0.58) ***, ###	0.38 (0.17–0.64)	0.34 (0.15–1.18)	0.29 (0.16–0.58) ***, #	0.41 (0.29–0.60)	0.39 (0.25–0.65)	0.22 (0.13–0.34) ***, ###, $	0.41 (0.23–0.81)	0.33 (0.20–0.56) *	0.32 (0.21–0.52) ***, #, §
**B- and T-cell cytokines**												
IL-2 (pg/mL)	8.91 (2.91–15.49)	8.51 (2.61–22.81)	8.69 (1.13–16.60)	10.76 (3.05–15.49)	10.01 (4.65–22.81)	9.71 (5.44–16.60)	8.25 (6.22–15.33)	7.55 (4.65–9.67) $$	6.99 (3.86–11.79) §	8.96 (3.31–14.62)	7.55 (2.61–14.67) *, $	8.25 (5.43–16.31)
IL-32 (pg/mL)	19.93 (1.05–72.57)	19.23 (0.57–88.09)	13.07 (1.050–77.75) #	19.93 (1.05–64.78)	17.23 (0.57–75.16)	14.51 (1.05–42.07)	9.78 (3.00–42.72) §	14.54 (2.39–45.18)	3.90 (2.39–45.18)	31.62 (3.00–72.57)	38.58 (3.00–88.09)	22.98 (3.00–77.75) $, §§
IL-34 (pg/mL)	109.10 (20.00–669.90)	151.60 (39.61–714.80)	106.40 (20.00–613.70)	77.97 (20.00–423.10)	86.42 (20.00–273.40)	120.60 (20.00–282.60)	51.90 (51.90–214.20)	120.60 (39.61–374.50)	51.90 (39.61–342.00)	151.40 (20.00–669.90)	179.00 (39.61–714.80) $	123.20 (20.00–613.70)
IL-35 (pg/mL)	287.1 (127.7–612.0)	298.90 (145.8–661.2)	302.30 (145.8–744.6)	302.6 (141.5–45.1)	304.6 (191.5–661.2) *	302.6 (175.1–390.9)	275.8 (145.8–375.9)	283.4 (181.6–348.6)	263.7 (181.6–523.5)	319.3 (198.3–612.0)	287.9 (153.8–425.1) *	314.4 (198.3–744.6)
TSLP (pg/mL)	35.45 (16.24–94.25)	34.52 (16.20–180.60)	37.76 (19.29–129.20)	38.04 (16.24–60.56)	36.80 (23.19–102.80)	35.45 (23.19–55.41)	34.10 (16.62–64.78)	33.92 (20.25–66.62) *	38.04 (23.86–68.44) *	36.28 (23.92–94.25)	33.15 (16.20–180.60)	39.79 (19.29–129.20)
**Cytokine receptors**												
gp130 (ng/mL)	39.20 (22.38–110.91)	39.09 (20.77–85.96)	33.87 (14.53–93.61) *, ###	36.32 (22.38–96.58)	33.25 (20.77–82.75)	31.27 (19.16–64.04) *	40.43 (27.99–53.59)	42.35 (23.36–58.34)	32.65 (14.53–43.47) ***, ###	39.19 (23.56–110.91)	44.26 (27.08–85.96) $	38.30 (21.82–93.61) $, §§
sIL-6Rβ (ng/mL)	6.90 (1.38–17.04)	6.87 (1.49–32.87)	5.70 (1.49–32.61) ###	6.00 (1.38–12.69)	6.13 (1.49–13.49)	5.79 (1.49–11.39) *	7.16 (4.26–11.50)	6.92 (3.72–12.07)	4.66 (2.03–8.96) ***, ###	7.09 (3.58–17.04)	7.28 (4.15–32.87)	6.19 (3.57–32.61) #, §
**Class II Cytokines**												
IL-10 (pg/mL)	20.91 (9.82–41.22)	27.07 (15.65–50.28) ***	23.11 (14.05–40.45) ***, #	21.53 (10.45–27.60)	27.11 (18.45–37.36) ***	26.74 (17.63–34.19) ***	20.05 (11.66–32.12)	26.04 (17.21–50.28) ***	24.14 (16.75–30.79) ***	21.45 (9.82–41.22)	27.39 (15.65–39.69) **	26.69 (14.05–40.45) **
IL-19 (pg/mL)	64.37 (41.37–86.96)	64.85 (43.86–110.10)	61.12 (38.71–108.50)	61.01 (41.37–74.26)	59.96 (49.46–110.10)	57.45 (40.07–69.61)	64.06 (49.46–86.96)	69.78 (55.35–102.20) §	64.06 (41.15–86.03) ##	66.23 (47.17–82.08)	58.65 (43.86–76.93) *	61.12 (38.71–108.50) $
IL-20 (pg/mL)	56.86 (24.68–97.81)	48.78 (23.72–85.47) **	48.75 (26.67–91.63) *	60.92 (34.35–97.81)	52.80 (38.84–85.47)	48.75 (38.84–87.58)	56.04 (24.68–74.35)	47.53 (28.66–70.68)	48.14 (26.67–81.73)	60.31 (34.37–81.36)	50.59 (23.72–77.26) *	49.69 (34.37–91.63)
IL-22 (pg/mL)	37.53 (8.73–87.88)	37.87 (10.18–90.53)	37.53 (8.73–123.00)	37.53 (8.73–87.88)	40.81 (23.77–90.53) *	38.21 (25.18–99.84)	37.53 (20.90–55.76)	37.19 (26.59–50.64)	32.12 (8.73–60.09)	39.00 (14.96–80.02)	35.94 (10.18–72.16)	39.71 (17.38–123.00)
IL-26 (pg/mL)	106.6 (36.9–323.9)	136.1 (74.3–308.9) ***	145.2 (67.1–254.0) ***	98.8 (53.4–323.9)	132.1 (98.7–308.9) ***	134.3 (98.76–254.0) ***	122.5 (57.9–243.1)	148.8 (101.0–218.4) *	146.5 (82.7–178.7) **	111.1 (36.9–173.2)	133.4 (74.3–157.4) *	140.5 (67.1–178.7) ***, ##
IL-28A/IFNλ2 (pg/mL)	11.53 (1.44–98.33)	13.93 (1.80–122.10)	12.80 (1.80–80.05)	33.21 (1.44–84.99)	30.75 (1.80–89.94)	52.08 (1.80–73.46)	29.11 (1.80–98.33)	18.71 (1.80–85.21)	1.92 (1.80–80.05) $$	1.80 (1.80–60.72)	1.80 (1.80–122.10)	1.80 (1.80–68.12) $$
IL-29/IFNλ1 (pg/mL)	38.95 (5.20–72.92)	38.95 (10.05–78.05)	35.51 (5.20–92.96)	44.89 (5.20–72.92)	45.44 (10.05–78.05)	37.89 (5.20–92.96)	38.95 (10.05–53.24)	38.95 (10.05–53.24)	33.03 (10.05–56.88)	35.33 (12.84–58.46)	34.22 (12.84–60.68) $	35.33 (12.84–62.88)
IFNα2 (pg/mL)	49.55 (32.01–95.10)	48.03 (27.81–117.50)	46.56 (25.12–137.00)	55.65 (32.01–79.95)	50.33 (38.58–117.50)	50.33 (38.00–68.60)	46.56 (30.24–54.14)	46.56 (30.24–54.14) $	40.37 (25.12–77.03)	51.13 (34.13–95.10)	45.02 (27.81–76.17) *, $	48.06 (27.81–137.00)
IFNβ (pg/mL)	16.92 (8.33–35.75)	17.23 (2.39–41.13)	14.72 (3.51–34.41)	16.00 (10.90–35.75)	17.16 (8.40–41.13)	13.44 (3.51–27.56)	16.92 (11.28–25.93)	16.92 (9.88–25.03)	14.09 (8.48–25.93)	21.20 (8.33–32.41)	21.37 (2.39–29.08)	19.81 (8.82–34.41)
IFNγ (pg/mL)	54.74 (17.95–87.71)	49.91 (22.38–152.30)	46.95 (13.35–113.7)	62.22 (21.19–84.15)	60.35 (29.97–152.3)	57.37 (32.07–88.83)	49.91 (17.95–72.97)	48.51 (22.38–62.22)	43.02 (13.35–96.49)	62.22 (38.99–87.71)	43.78 (22.44–94.78) *	46.95 (29.19–113.70)
**Matrix metalloproteinases and tissue remodeling factors**												
MMP-1 (ng/mL)	0.76 (0.26–3.00)	0.67 (0.05–3.46)	0.81 (0.24–2.71)	0.80 (0.29–2.47)	0.97 (0.05–2.56)	0.80 (0.24–2.22)	0.77 (0.44–2.22)	0.61 (0.24–2.90)	0.82 (2.90–1.84)	0.73 (0.26–3.00)	0.55 (0.18–3.46)	0.62 (0.31–2.71)
MMP-2 (ng/mL)	20.69 (6.34–41.00)	17.76 (3.96–153.49)	14.37 (3.19–49.31) **, #	19.94 (7.88–35.26)	17.46 (6.22–153.49)	16.95 (7.51–49.31)	27.45 (6.34–40.25)	22.06 (3.96–121.13)	9.23 (3.19–29.30) ***, #, $$	17.43 (11.99–41.00)	17.32 (5.84–39.84)	17.66 (6.28–37.10) §
MMP-3 (ng/mL)	6.66 (1.83–27.98)	7.37 (2.04–188.25)	11.69 (3.33–450.14)	7.27 (2.52–16.93)	7.50 (3.16–49.20) ***	15.88 (3.89–38.10)	7.49 (1.83–16.67)	6.58 (3.26–31.28) *	11.97 (4.59–46.64) #	5.13 (2.07–27.97)	7.08 (2.04–188.25)	6.13 (3.33–450.14)
Chit3like1/YKL-40 (ng/mL)	6.24 (1.76–24.50)	13.96 (2.95–96.45) ***	9.05 (2.48–42.36) ***, ###	5.39 (1.76–24.50)	12.97 (4.31–30.34) ***	8.57 (2.48–21.47) ***, ###	6.42 (3.03–17.44)	14.94 (2.95–28.47) ***	8.52 (4.43–25.14) ***, ###	8.04 (2.76–23.10)	15.45 (4.38–96.45) ***	10.75 (4.00–42.36) ***, ###
**Bone markers**												
OC (ng/mL)	3.99 (1.15–7.82)	2.57 (0.67–7.45) ***	2.40 (0.58–7.10) ***	2.85 (1.15–6.60)	2.18 (0.78–4.69) **	2.16 (0.78–5.16) **	4.09 (1.98–7.78)	2.55 (0.67–7.45) *	1.98 (0.58–4.47) ***, #	4.55 (2.43–7.82)	3.63 (0.95–6.26) ***	3.25 (0.97–7.10) ***, $, §§
OPN (ng/mL)	45.10 (15.11–152.74)	64.70 (31.83–105.83) ***	64.09 (37.98–174.03) ***	40.50 (22.60–60.65)	57.86 (37.55–73.79) ***	63.44 (38.86–86.10) ***	49.17 (15.11–108.03)	69.79 (37.27–105.83) ***, $	61.42 (37.97–104.02) ***	47.07 (16.18–152.74)	68.63 (31.83–174.03) $	72.27 (38.34–174.03) *

Values are expressed as median (range). *: difference vs. T0 (*: *p* < 0.05; **: *p* < 0.01; ***: *p* < 0.001); #: difference vs. T1 (#: *p* < 0.05; ##: *p* < 0.01; ###: *p* < 0.001); $: difference vs. ALIF ($: *p* < 0.05; $$: *p* < 0.01); §: difference vs. TLIF (§: *p* < 0.05; §§: *p* < 0.01). ALIF: Anterior Lumbar Interbody Fusion; TLIF: Transforaminal Lumbar Interbody Fusion; LLIF: Lateral Lumbar Inter-body Fusion; CRP: C-reactive protein; sCD: soluble cluster of differentiation; IL: interleukin; PTX-3: pentraxin-3; APRIL: a proliferation-inducing ligand; BAFF: B-cell activating factor; LIGHT: homologous to lymphotoxin, exhibits inducible expression and competes with HSV glycoprotein D for binding to herpesvirus entry mediator, a receptor expressed on T lymphocytes; sTNFR: soluble tumor necrosis factor receptor; TWEAK: tumor necrosis factor-related weak inducer of apoptosis; TSLP: thymic stromal lymphopoietin; gp130: glycoprotein 130; sIL-6Rβ: soluble interleukin 6 receptor β; IFN: interferon; MMP: matrix metalloproteinases; CHI3L1: chitinase-3-like protein 1; OC: osteocalcin; OPN: osteopontin.

## Data Availability

Data supporting reported results can be found at doi:10.5281/zenodo.4724715.

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
