# Peer review of "Peri-Surgical Inflammatory Profile Associated with Mini-Invasive or Standard Open Lumbar Interbody Fusion Approaches"

_jcm, 2021, doi:10.3390/jcm10143128_

Round 1

Reviewer 1 Report

This is a sophisticated study on the detection of inflammatory markers in patients undergoing lumbar fusion. The authors found that MIS procedures (ALIF and LLIF) were associated with a reduced incidence of post-operative anemic status, faster recovery and enhanced pro-angiogenic stimuli compared with TLIF. Moreover, LLIF associated with an earlier activation of innate immune mechanisms than ALIF and TLIF.

Although the authors put a tremendous work, some methodological flaws exist. It is unclear why a real control group is missing to compare with.

What is also missing and would be of particular interest is clinical and radiological correlations with the studied markers. Only elevated numbers do not mean much.

There is no data after the 4th postoperative day. The authors provide an explanation but it not fully convincing.

The clinical perspective of the current findings is inadequately addressed.

Overall, the manuscript is too extended and difficult to follow. It is recommended to present only the most important data, avoiding repetition and duplication of the results, when possible.

Reviewer 2 Report

The aim of the study is worthy, but unfortunately the results are confusing and do not support any general hypothesis raised. The finding that blood loss and associated hematologic markers are more affected by open TLIF compared to ALIF or LLIF is hardly novel or of interest. However, open TLIF was not associated with increased postop inflammation, as one would assume the investigators hypothesized, most notably seen in the CRP results. In the conclusions, the investigators seem to praise minimally invasive procedures when  they are associated with increased inflammatory responses--like angiogenic factors, indicating this is some type of advantage. It is likely that postop inflammatory markers are also influenced by anesthetic regimen, pain, and other factors. I see no clear data in this study that minimally invasive approaches impact them in a consistent way, though the authors appear to believe their data indicates some superiority, which I don't feel is supported by the data.
